# Cytokines Stimulated by IL-33 in Human Skin Mast Cells: Involvement of NF-κB and p38 at Distinct Levels and Potent Co-Operation with FcεRI and MRGPRX2

**DOI:** 10.3390/ijms22073580

**Published:** 2021-03-30

**Authors:** Kristin Franke, Zhao Wang, Torsten Zuberbier, Magda Babina

**Affiliations:** 1Department of Dermatology, Venerology and Allergology, Charité-Universitätsmedizin Berlin, Corporate Member of Freie Universität Berlin and Humboldt-Universität zu Berlin, Charitéplatz 1, 10117 Berlin, Germany; kristin.franke@charite.de (K.F.); zhao.wang@charite.de (Z.W.); torsten.zuberbier@charite.de (T.Z.); 2Department of Dermatology, The Second Affiliated Hospital, Northwest Hospital, Xi’an Jiaotong University, Xi’an 710004, China

**Keywords:** mast cells, IL-33, skin, cytokines, signaling, p38, NF-κB, FcεRI, MRGPRX2, synergism

## Abstract

The IL-1 family cytokine IL-33 activates and re-shapes mast cells (MCs), but whether and by what mechanisms it elicits cytokines in MCs from human skin remains poorly understood. The current study found that IL-33 activates CCL1, CCL2, IL-5, IL-8, IL-13, and TNF-α, while IL-1β, IL-6, IL-31, and VEGFA remain unaffected in cutaneous MCs, highlighting that each MC subset responds to IL-33 with a unique cytokine profile. Mechanistically, IL-33 induced the rapid (1–2 min) and durable (2 h) phosphorylation of p38, whereas the phosphorylation of JNK was weaker and more transient. Moreover, the NF-κB pathway was potently activated, as revealed by IκB degradation, increased nuclear abundance of p50/p65, and vigorous phosphorylation of p65. The activation of NF-κB occurred independently of p38 or JNK. The induced transcription of the cytokines selected for further study (CCL1, CCL2, IL-8, TNF-α) was abolished by interference with NF-κB, while p38/JNK had only some cytokine-selective effects. Surprisingly, at the level of the secreted protein products, p38 was nearly as effective as NF-κB for all entities, suggesting post-transcriptional involvement. IL-33 did not only instruct skin MCs to produce selected cytokines, but it also efficiently co-operated with the allergic and pseudo-allergic/neurogenic activation networks in the production of IL-8, TNF-α, CCL1, and CCL2. Synergism was more pronounced at the protein than at the mRNA level and appeared stronger for MRGPRX2 ligands than for FcεRI. Our results underscore the pro-inflammatory nature of an acute IL-33 stimulus and imply that especially in combination with allergens or MRGPRX2 agonists, IL-33 will efficiently amplify skin inflammation and thereby aggravate inflammatory dermatoses.

## 1. Introduction

Mast cells (MCs), the principal effector cells of type-I allergic reactions, terminally mature and are maintained by MC-supportive niches in tissues, of which interleukin (IL)-33 can be a component. This member of the IL-1 family, that signals through the ST2/IL1RacP complex, has gained attention due to its potent effects on MCs [1,2,3,4,5,6]. IL-33 acts as a MC activator and component of the MC-supportive micromilieu, as exemplified by the fact that mice lacking an IL-33 receptor have lower numbers of MCs in several tissues, while provision of exogenous IL-33 increases the MC compartment [7,8].

IL-33 combines pro-inflammatory characteristics with a Th2-skewing potential [9,10] and is critically implicated in allergic disorders of the respiratory tract [11,12,13] as well as in food allergy [14,15,16]. In addition, aberrant IL-33 regulation is linked to chronic dermatoses like atopic dermatitis, psoriasis, and urticaria, whereby several in vivo models proved IL-33′s contribution to the respective pathologies [17,18,19]. Moreover, IL-33 also acts acutely as an “alarmin” when released by dying cells [20,21], and it was found to be the most potent alarmin among several entities to activate MCs [22]. Accordingly, the direct intradermal application of IL-33 results in acute and chronic skin inflammation [23].

Degranulation is a key function of MCs, characterized by the rapid release of highly active mediators that elicit hypersensitivity reactions. In addition to the canonical high affinity IgE receptor (FcεRI)-induced activation [24,25,26], MAS-related G protein-coupled receptor X2 (MRGPRX2) was recently uncovered as a further stimulatory route for MCs in human skin [27,28]. MRGPRX2 is almost confined to skin MCs, where it acts as the receptor of various ligands, including widely investigated entities, namely compound 48/80 (c48/80) and substance P (SP) [27,29,30,31].

Both receptor systems are largely equipotent in vitro and in vivo [32,33,34,35,36,37]. In contrast to the degranulation-competent receptors, the ST2/IL1RacP complex does not degranulate MCs directly when triggered by IL-33 [38,39,40,41,42,43,44,45]. IL-33 can however prime for enhanced granule discharge when combined with stimulation via the canonical allergic route in different types of MCs [42,43,45,46,47,48]. We recently reported that degranulation elicited by the novel pseudo-allergic route is likewise supported by IL-33 priming in skin MCs [47], while no such effect was noted in peripheral blood-derived mast cells (PBMCs) [43].

In addition to degranulation, MCs are efficient producers of cytokines, though the precise pattern will depend on the MC subset. For example, in contrast to most other MC types [41,49], FcεRI aggregation of skin MCs does not elicit IL-6, while it increases other pro-inflammatory and Th2-related cytokines, including tumor necrosis factor (TNF)-α, IL-8, IL-5, IL-13, and IL-31 [50,51,52,53]. Moreover, the CC chemokines CCL1 and CCL2 (but not CCL3, 4, and 5) can be potently activated in cutaneous MCs [50,54,55,56,57].

Contrary to degranulation, IL-33 has been reported to elicit cytokines on its own in several types of murine and human MCs [14,38,39,40,58,59,60,61,62,63,64,65], while little is known for skin MCs. Of note, not only the cytokine entities produced but also the mechanisms underlying their production differ across MC types (as will be further detailed in the Section 3).

Considering the foregoing reports, we sought to reveal whether and by what mechanisms skin MCs respond to IL-33 with cytokine generation. We report that selective cytokines are indeed effectively triggered, while other frequently studied entities (such as IL-6 and VEGFA (vascular endothelial growth factor A)) are not regulated by IL-33 in skin MCs. Mechanistically, the activation of three major cascades can be observed, namely p38, c-Jun N-terminal kinase (JNK), and nuclear factor kappa-light-chain-enhancer of activated B cells (NF-κB), and we identify their differential contribution to cytokine generation with a rank order of NF-κB > p38 > JNK. Interestingly, while NF-κB is key to cytokine mRNA induction, p38 seems to mainly act post-transcriptionally. Finally, we determine that IL-33 synergizes with classical stimulation by FcεRI aggregation or the newer MRGPRX2-dependent route to enhance cytokine responses in a greater than additive manner. By creating a cytokine-conducive milieu, IL-33 likely re-shapes MC functions in the cutaneous environment and can thereby contribute to skin inflammation and disease.

## 2. Results

### 2.1. IL-33 Elicits Cytokine Production in Skin MCs

We analyzed the pattern of multiple cytokines expressed by skin-derived MCs at baseline or upon FcεRI-aggregation [50,51,52]. Of these, IL-33 potently increased TNF-α, CCL1, CCL2, IL-8, IL-5, and IL-13, while IL-1β, IL-6, and VEGFA were unaffected by IL-33, and IL-31 was below detection.

While Figure 1a shows the FC (fold change) over control, and thus the degree of induction by IL-33, Figure 1b gives the actual expression (normalized against beta-actin as housekeeping gene) to allow for broad comparisons of cytokine abundance (see also Appendix A for normalization against Cyclophilin B and GAPDH). Both figures together illustrate that the CCL chemokines were already produced considerably at baseline despite robust enhancement by IL-33. On the other hand, while IL-33 had a potent effect on IL-13; this was mainly due to its marginal expression in control cells (Figure 1b and Appendix A).

Since CCL1, CCL2, IL-8, and TNF-α were consistently regulated by IL-33, we next quantified their release using ELISA (Figure 1c). In fact, IL-33 increased the selected cytokines by a factor between 4 (for CCL2) and 74 (for TNF-α) over baseline. We conclude that IL-33 efficiently stimulates selected cytokines in human skin MCs.

### 2.2. IL-33 Prompts the Phosphorylation of p38, JNK, and Potently Activates the NF-κB Pathway in Cutaneous MCs

We previously reported that IL-33 elicits p38 activation in skin MCs, coupled with weaker JNK, while signals for other phospho-kinases were inconsistent and weak on comparison with stem cell factor (SCF) [44]. Here, we studied the phosphorylation kinetics of p38 and JNK in greater detail. As depicted in Figure 2a,b, p38 experienced rapid and long-lasting activation starting already at 1–2 min (i.e., the shortest time periods selected), reaching a maximum between 8–30 min, and subsiding thereafter. However, the respective signal was still above baseline even after 2 h. In contrast, JNK activation was transient, starting at 5, peaking at 15, and almost vanishing by 30 min.

We turned our attention to the NF-κB route not previously investigated in skin MCs. NF-κB is frequently, yet not invariably activated by IL-33 [62,66,67,68], suggesting cell-type dependence. We found very rapid (1–2 min) phosphorylation of pp65 at Ser536, which was maintained for ≈8 min and progressively but slowly declined thereafter. After 2 h, the signal was still not completely abolished vis-à-vis control (Figure 2a,d). Occasionally slight differences in the molecular weight of p65 were observed, potentially resulting from phosphorylation(s) and/or other types of post-translational modifications. The near complete degradation of IκBα was elicited, as well, starting at 8 min, peaking at 12–15 min followed by re-synthesis, yet again without reaching its original level by 2 h (Figure 2a,e).

We tested whether p38 and JNK could feed into the NF-κB route as occasionally reported [69], but found that this was not the case. In fact, the IL-33-triggered degradation of IκBα was comparable in cells pre-exposed to a p38 inhibitor (p38i) or JNK inhibitor (JNKi) (Figure 3a). The same result was seen for pp65, which remained unaffected by p38 or JNK suppression. However, p38i interfered with the phosphorylation of its own target, and this was similarly reproduced for JNK, i.e., the antagonization of JNK phosphorylation by JNKi.

We finally assessed whether IκBα degradation was associated with p65 and/or p50 translocation to the nucleus. Consistently with IκBα degradation, nuclear p65 increased gradually from 0 to 30 min (Figure 3b). Concurrently, there was a slight decrease in cytoplasmic p65. An increase in nuclear p50 was likewise noted, and this was accompanied by a reduction of nuclear p105, suggesting intranuclear processing by the proteasome [70,71]. Overall, the IL-33-elicited increase in the nucleus was more pronounced for p65 than for p50.

We conclude that in addition to JNK and p38, IL-33 potently activates the canonical NF-κB pathway in skin MCs.

### 2.3. Cytokine Generation by IL-33 Chiefly Depends on NF-κB and p38: Differences between RNA and Protein Induction

We used specific pharmacological inhibitors of the above pathways to study their involvement in IL-33-stimulated cytokine generation. The induction of TNF-α and CCL2 mRNA was only inhibitable by Bay 11-7082 and therefore chiefly driven by NF-κB, while p38 or JNK made no significant contribution (Figure 4a). Interestingly, this was different for the CCL1 transcript, which was not only sensitive to Bay 11-7082 but also to the p38 inhibitor SB203580 (Figure 4a) and the IL-8 transcript, whose induction was also countered by the JNK inhibitor SP600125.

Addressing the actual cytokine release, we verified that all entities were strongly suppressed in the presence of Bay 11-7082 as expected, since transcript induction is a prerequisite to protein generation (Figure 4b). Surprisingly, however, interference with p38 blocked cytokine release nearly as efficiently as Bay 11-7082 (Figure 4b). Even JNKi had a slightly suppressive effect, not only in the case IL-8 but also for TNF-α. Therefore, the regulation of the secreted protein product is more complex requiring support from different participants with the average rank order being NF-κB > p38 > JNK. In contrast, canonical signaling through NF-κB is responsible for much of the cytokine transcripts induced by IL-33 irrespective of p38 (with the exception of CCL1) and JNK activity (with the exception of IL-8).

### 2.4. IL-33 Efficiently Synergizes with FcεRI and MRGPRX2 to Elicit Cytokine Production

MRGPRX2 is highly expressed by skin MCs as shown by us and others [27,29,30,31,50]. RT-qPCR, flow-cytometry, and immunofluorescence were employed to confirm MRGPRX2 expression in our MC model (Appendix A). Skin MCs were compared to cells of the human mast cell line HMC-1 by RT-qPCR, confirming high expression in skin MCs, while HMC-1 displayed minute expression at best [72]. We also recently found that MRGPRX2 stimulation by c48/80 and SP can trigger cytokines in MCs of skin origin, albeit at lower levels compared to the allergic route [54]. IL-33 can synergize with FcεRI in several MC subsets [39,41,45,46,58,73] and this prompted interest in whether synergy between IL-33 and other receptor systems also occurs in MCs from human skin. We were especially interested in whether IL-33 and MRGPRX2 can reinforce each other’s ability to induce cytokines, and if so, at what level.

We found that the combined treatment elevated cytokine transcripts over individual stimuli (Figure 5a and Appendix A). An even more accentuated effect was obtained for TNF-α and IL-8 protein (Figure 5b).

To determine whether effects were additive or synergistic, we added the values obtained for individual treatments and compared the results with the actual measured amounts (i.e., stimulus 1 + 2 combined/stimulus 1 + stimulus 2). The calculated “synergism factors” are displayed in Table 1, whereby 1 signifies a merely additive effect, while numbers well above 1 denote synergism. In the context of IgER-CL, IL-33 had a rather additive effect on cytokine mRNA expression (except for IL-8). For example, TNF-α mRNA simultaneously triggered by FcεRI-aggregation and IL-33 did not exceed the sum of the individual effects, while the corresponding factor was 5.9 for IL-8 (Table 1). IL-8 also turned out to be the most potently elevated transcript in the setting of MRGPRX2 stimulation by either compound 48/80 (c48/80) or substance P (SP), possibly resulting from the involvement of JNK in its transcription (pJNK is detected upon IL-33, but not or weakly after FcεRI aggregation or MRGPRX2 ligation [54]). The least synergism was found for TNF-α mRNA. The picture was quite different at the protein level. Synergism factors for both cytokines measured by ELISA, i.e., TNF-α and IL-8, exceeded the corresponding factors achieved by their transcripts. The most striking example was TNF-α triggered via FcεRI (8.1 versus 1.1). With the exception of the latter, the datasets viewed in aggregate indicate that the synergistic behavior tends to be more pronounced for the pseudo-allergic/neurogenic compared to the allergic route.

We conclude that IL-33 does not only stimulate cytokines on its own, but also creates a cytokine-favoring micromilieu for other crucial receptors of skin MCs, namely FcεRI and MRGPRX2. The results for combined stimulations further highlight the importance of post-transcriptional events for cytokine generation and its support by IL-33, pointing towards a crucial role of p38 in driving post-transcriptional synergism.

## 3. Discussion

As components of innate immunity as well as bridges between innate and adaptive immune functions, MCs are ideally localized in tissues to act as sentinels that organize antimicrobial and anti-parasitic defenses [74,75]. On the other hand, MCs can also trigger hypersensitivity reactions upon exposure to exogenous antigens, drugs, venoms, or endogenous alarmins released by damaged or inflamed tissues. Among the latter category, IL-33 is arguably the most significant entity to affect MCs [22]. In fact, IL-33 can elicit a large spectrum of inflammatory (and to a certain degree also anti-inflammatory) responses in MCs, the precise nature of which will however depend on the MC subset and conditions.

How MCs in skin, arguably the most abundant MC population in the body [76], are impacted and remodeled by IL-33 has only recently begun to be elucidated. In particular, prolonged exposure to IL-33, mimicking the chronically inflamed skin, was found to reinforce MCs numerically through accelerated cell-cycle progression and reduced cell death. However, it simultaneously attenuated FcεRI and MRGPRX2 expression and degranulation elicited by their activation, but increased levels of the major allergic mediator histamine, thereby exhibiting the character of a double-edged sword [44,47]. Conversely, an acute burst of IL-33 was found to support both FcεRI- and MRGPRX2-dependent degranulation in a p38-dependent manner [47].

In addition to preformed mediators, MCs synthesize cytokines de novo that are secreted in a timely shifted fashion and contribute to chronic inflammatory diseases [25,49,77,78]. Consistent with reports on other MC subsets, we first confirm that cytokines and chemokines are elicited by IL-33 in cutaneous MCs, but also find that patterns across MC types differ. While CCL1, CCL2, IL-5, IL-8, IL-13, and TNF-α were induced (or enhanced vis-à-vis an already prominent baseline) in cutaneous MCs, IL-33 did not affect IL-1β, IL-6, IL-31, or VEGFA.

The lack of effect of IL-33 on IL-6, IL-31, and VEGFA in dermal MCs differs from reports on other MC subsets, as especially IL-6 is a frequent target of IL-33 in MCs [38,41,58,63,64]. In addition to IL-6, VEGFA was positively affected by IL-33 in LAD2 MCs [17] and human lung MCs [45], IL-31 in HMC-1 cells [79] and IL-1β in cord blood-derived mast cells (CBMCs) [80], highlighting that the MC subset plays a crucial part in dictating the cytokine pattern. In addition, while IL-13 was also upregulated by IL-33 in our study (Figure 1a), its overall expression was low (Figure 1b and Appendix A) in contrast to IL-33-stimulated CBMCs and LUVA MCs [38,81].

However, the findings for IL-33-triggered activation correspond with other types of activation of skin MCs. For example, IL-6 expression is low in skin MCs (especially when compared to other immune cells, [50,55]) and unchanged by FcεRI-aggregation [51,52,53], and while VEGFA is highly expressed in skin MCs, it is likewise barely affected by stimulation via FcεRI [50]. We recently determined that cytokines generated in response to MRGPRX2 activation closely match those induced by FcεRI aggregation (with the exception of IL-31) [54], suggesting that the profile a subset generates may be more strongly dictated by its nature than by the stimulus. The present study extends this concept to IL-33. The combined data suggest that chromatin accessibility at the relevant loci prior to activation may be differently organized across MC subsets depending on the needs dictated by the respective microenvironment.

In response to ligand binding, the ST2/IL-1RacP complex will initiate a sequence of phosphorylation and ubiquitination events starting with the recruitment of the adapter MyD88 (myeloid differentiation primary response gene 88) through its TIR domain (toll/interleukin-1 receptor), followed by members of the IRAK family (interleukin-1 receptor–associated kinase), the E3 ubiquitin ligase TRAF6 (TNF receptor-associated factor 6), and eventually TAK1 (TGF-β-activated kinase 1) [68,74,82,83]. TAK1 is the starting point of several cascades, among which are the stress-associated kinases p38 and JNK as well as NF-κB. These downstream events are commonly activated in numerous cell types, although strength and relative proportions are cell-type specific. For example, a dominance of p38 was reported in oligodendrocyte precursor cells [84], the heart [85], memory Th2 cells [86], and ST2^+^ Tregs [87], while NF-κB was the most important pathway in hepatocytes [88]. In myeloid cells, to which MCs belong, IL-33 seems to preferentially induce both p38 and NF-κB [89,90,91,92]. Yet other cell types such as bronchial epithelial cells ERK1/2 can dominate in the absence of p38 and JNK activation [93].

Without addressing the NF-κB module, we recently reported for human skin MCs that IL-33 leads to the robust phosphorylation of p38 and slight phosphorylation of JNK. p38 was involved in the upregulation of histidine decarboxylase and histamine [44] as well as in IL-33-supported degranulation [47], while JNK activity seemed to underlie the IL-33 mediated downregulation of MRGPRX2 (potentially acting as a negative feedback mechanism) [47]. Here, we investigate the detailed kinetics of these phospho-events, determining that pJNK appears transiently, while pp38 is rather persistent, starting rapidly and lasting for at least 2 h.

The significance of p38 has been widely illustrated in different MC subsets. In murine peritoneal mast cells, p38 contributed to IL-33-driven cell cycle progression [8]. In CBMCs, IL-33 induced the phosphorylation of pERK and pJNK, but the most pronounced event was related to pp38 [39], while in LAD2 cells pp38 and pJNK were detected, but the pJNK signal was rapidly diminished (as in our study) and pp38 was more persistent [62]. In a study with bone marrow-derived mast cells (BMMCs), the stimulation of NF-κB and p38 were also the most pronounced events [94]. In HMC-1 and murine BMMCs various signaling events were noted, but several of those required complex formation between c-Kit, IL-33R, and IL-1RAcP, whereas pp38 (and pp65) were elicited largely independently of KIT [59]. In skin MCs most IL-33 signaling seems KIT-independent since cascades are well distinguishable between SCF/KIT and IL-33/ST2 [44].

Whether NF-κB forms part of the signaling modules induced by IL-33 in skin MCs has not been previously addressed. We now report on the potent activation of the canonical NF-κB cascade, including IκBα degradation supplemented by increased nuclear abundance of p50 and more so, p65. We also find rapid and durable p65 phosphorylation at Ser563, an event that precedes IκBα degradation. Some p50 and p65 can be found in the nucleus at baseline in accordance with the literature [95], but p65 experiences considerable translocation from the cytoplasm upon IL-33 stimulation, closely matching the kinetics of IkBα degradation. The increase in nuclear p50 accumulation may stem, at least in part, from the proteasomal degradation of its precursor p105 in the nucleus because of the concomitant decrease of the latter [70,71].

While NF-κB activation by IL-33 is commonly observed, its precise components can vary among MC subtypes. One study described the preferential translocation of p50 in BMMCs in the absence of p65 translocation [60]. Another study showed that for LAD2 cells IL-33 activates p65 and IκBα phosphorylation, but the latter event appeared unexpectedly late as it was most pronounced at 1–4 h after stimulation [96].

We also demonstrate that p38 and JNK do not modulate activation of the NF-κB pathway and are thus independent events. However, p38 seems to be partially responsible for its own phosphorylation, and JNK likewise shows this autocatalytic behavior in skin MCs. This is in accordance with the literature, whereby p38-mediated p38 activation represents the non-canonical pathway of its activation [97,98,99]. For example Canovas and colleagues state that besides the canonical phosphorylation of p38 by its specific MAP kinase kinases “in the non-canonical pathways, activation is triggered by autophosphorylation of p38α”. [98]. However, the occurrence of these phenomena is cell-type dependent. In murine BMMCs for example, p38 inhibition did not inhibit its phosphorylation (while it did block cytokine responses) [100]. In contrast, in LAD2 cells, pJNK was slightly inhibited by JNKi, while pp38 was completely suppressed by p38i [62] resembling our study and suggesting that there may be differences between mouse and man in this regard.

We were interested to associate the above cascades to cytokine expression and release. At the mRNA level, the induction of the selected entities strongly depended on NF-κB, in accordance with the existence of potent NF-κB binding sites in their promoters [101]. In contrast, p38 or JNK made only some cytokine-selective contributions (p38 for CCL1, JNK for IL-8). Surprisingly, we found that the actual cytokine release depended on additional pre-requisites, as it was effectively antagonized not only by the NF-κB inhibitor (which suppressed transcript induction), but also by the perturbation of p38. Moreover, JNK contributed to the release of IL-8 (perhaps mainly transcriptionally) and TNF-α (probably post-transcriptionally). The clear distinction between IL-33-induced cytokine mRNA versus protein has, to our knowledge, rarely been discussed previously, even though several reports showed differences between the two levels of regulation. For example, the IL-33-induced FC for CCL2 was greater at protein than at mRNA level in LAD2 cells [62]. Another study found that the IL-33-induced activation of NF-κB was not or even positively affected by resveratrol, while cytokine production (e.g., of TNF-α) was simultaneously inhibited in a p38-dependent manner [63], suggesting that NF-κB-independent events may be dominant in cytokine regulation in specific circumstances. Additionally, a recent paper investigated the differential regulation of a number of cytokines on mRNA versus protein level in PBMCs by comparing IL-33 with the canonical allergic pathway [102]. One result was that even though the FC for CCL1 mRNA upon IgER-CL was higher (~3400) than for IL-33 (~400) the resulting protein release was greater for IL-33. Contrary to CCL1, the greater FC for IL-8 between IgER-CL and IL-33 on mRNA level (~1100 versus ~95) was reproduced on the protein level, implying cytokine-dependent peculiarities. For the four entities investigated by our study, however, the positive impact of p38 at the level of secreted products was largely uniform. Previous literature supports the involvement of p38 in IL-33 triggered cytokines, though mRNA and protein levels have rarely been compared. For example, IL-33-mediated IL-8 production by CBMCs was markedly reduced by SB203580 [39], while CCL2 release was likewise sensitive to p38 inhibition in LAD2 cells [62].

Pinpointing the modes of operation of p38 in post-transcriptional cytokine regulation will require future efforts. However, the first clues come from previous literature. For example, TNF-α (and IL-6) mRNA stability and translation have been found to depend on p38 in LPS-triggered BMMCs by a process requiring members of the 12-O-tetradecanoylphorbol-13-acetate (TPA) inducible sequence 11 (TIS11) family other than p38-MAPKAP kinase 2 (MK2)-tristetraprolin (TTP)) [103].

We finally demonstrate that IL-33 priming bestows a generally increased secretory competence on skin MCs at the level of degranulation, as reported previously [47], but even more so at the level of cytokines, as shown herein. Synergism between IL-33 and FcεRI is well established and applies to most MC types [39,41,45,46,58,64,65]. We confirm that the cross-talk between both receptor systems is efficient in skin-derived MCs, and interestingly find that it may mainly proceeds at the post-transcriptional level, especially in the case of TNF-α (Table 1).

Conversely, little is known about IL-33′s potential to co-operate with the newly uncovered MC-specific G-protein coupled receptor MRGPRX2. While several studies reported on potent synergy between SP and IL-33 in driving cytokine responses (such as TNF-α, IL-1β, and VEGFA) [17,80,96] the authors focused on SP’s capacity to signal via NK-1 (neurokinin-1) in LAD2 cells, another SP receptor, revealing a physical interaction between NK-1 and ST2 in these cells [80,96].

Conversely, it is now firmly established that in skin MCs, SP signals via MRGPRX2 as demonstrated by MRGPRX2 knockdown experiments [27,29,104]. This is further corroborated by the fact that SP shows extensive functional overlaps with c48/80, the latter representing the canonical and arguably most potent MRGPRX2 agonist [35,105,106]. Accordingly, degranulation responses correlate almost perfectly between c48/80 and SP in skin MCs [32] and are similarly regulated by extracellular cues like SCF, IL-33, and TSLP [32,47,107]. Here, we not only find co-operation between MRGPRX2 and IL-33, but also demonstrate that it is at least as potent or even more pronounced than for the FcεRI/IL-33 pair (Table 1). In addition, the preferential level at which IL-33 synergizes with the receptors may differ between FcεRI and MRGPRX2, since MRGPRX2-triggered cytokine transcripts are more potently supported by IL-33 co-stimulation, or put differently, transcriptional co-operation is higher for MRGPRX2 than for FcεRI when viewed across all cytokines together. It remains to be investigated whether this is related to the weaker cytokine production elicited via MRGPRX2 perhaps as a consequence of the distinct Calcium (Ca^2+^) responses between FcεRI and MRGPRX2, as nicely elaborated by Gaudenzio and colleagues [36]. In fact, the Ca^2+^/Calcineurin/nuclear factor of activated T-cells (NFAT) pathway is viewed as crucial for the synergy between FcεRI and IL-33 [41].

In summary, our study demonstrates that IL-33 generally fosters the cytokine producing apparatus of skin MCs through operation at different levels. IL-33 thus enables alarmin-induced and assists in allergic as well as pseudo-allergic/neurogenic inflammation in the cutaneous environment, potentially aggravating dermatoses like AD, urticaria, and psoriasis.

## 4. Materials and Methods

### 4.1. Cells and Treatments

MCs were isolated from human foreskin tissue as described [32,47,51]. To achieve sufficient cell numbers, each mast cell preparation/culture originated from several (2–10) donors, as routinely performed in our lab [27,44,47,108,109,110]. The skin was obtained from circumcisions, with written, informed consent of the patients or legal guardians and approval by the university ethics committee (protocol code EA1/204/10, 9 March 2018). The experiments were conducted according to the Declaration of Helsinki principles. Briefly, the skin was cut into strips and treated with dispase (Boehringer-Mannheim, Mannheim, Germany) at 0.5 mg/mL at 4 °C overnight, the epidermis was removed and the dermis finely chopped, and then digested with 1.5 mg/mL collagenase (Worthington, Lakewood, NJ, USA), 0.75 mg/mL hyaluronidase (Sigma, Deisenhofen, Germany), and DNase I at 10 µg/mL (Roche, Basel, Switzerland). The cells were filtered stepwise from the resulting suspension (100 µm and 40 µm strainers). To further purify the MCs, anti-human c-Kit microbeads and an Auto-MACS separation device were used (both from Miltenyi-Biotec, Bergisch Gladbach, Germany), giving rise to 98–100% MC preparations. The purity of the isolated skin mast cells was verified by FACS (double staining for KIT/FcεRI) and acidic toluidine blue staining (0.1 % in 0.5 N HCl), as described [111,112].

MCs were cultured in the presence of SCF (100 ng/mL) and IL-4 (20 ng/mL), freshly provided twice weekly when cultures were re-adjusted to 5 × 10^5^/mL. MCs were automatically counted by CASY-TTC (Innovatis/Casy Technology, Reutlingen, Germany) [27,108,113].

Experiments were performed 3–4 d after the last addition of cytokines and cells were deprived of growth factors (GFs) and FCS (minimal medium) for 16 h prior to stimulation for downstream experiments. Each experiment was performed several times and for each time an individual culture was used (corresponding to a “dot” in the dot plots displayed in the figures). For inhibition studies, cells were pre-incubated with SB203580 (p38-Inhibitor; 20 µM), SP600125 (JNK Inhibitor; 20 µM), or BAY11-7082 (NF-κB inhibitor; 10 µM), all from Enzo Life Sciences, Germany for 15 min. IL-33 was purchased from PeproTech (Hamburg, Germany) and applied in a concentration of 20 ng/mL, as described [47]. Inhibitor concentrations were selected based on literature, IC50 values, and previous own studies. The concentrations of AER-37 (anti-FcεRIα-Ab; eBioscience, San Diego, CA, USA), c48/80 (Sigma, Steinheim, Germany), and SP (Bachem, Budendorf, Switzerland) were as follows: 0.1 µg/mL, 10 µg/mL, and 30 µM, respectively, in accordance with our previous studies [27,32,54,114,115,116,117].

### 4.2. Reverse Transcription-Quantitative PCR (RT-PCR)

MCs (at 5 × 10^5^ cells/mL) in minimal medium were treated with the according inhibitors for 15 min prior to IL-33 addition. For combinations of additional stimuli with IL-33, cells were pre-treated with IL-33 for 15 min followed by the addition of AER-37, c48/80, or SP and incubated for another 90 min. RT-qPCR was performed using pre-optimized conditions [109,112,118]. Primer pairs are summarized in Table 2. The 2^-ΔΔCT^ method was performed to quantify the relative expression levels of the target genes to three reference genes [47]. A “synergism factor” was calculated for each combined treatment of IL-33 plus IgER-CL or MRGPRX2-ligand, as described under Section 4.3.

### 4.3. ELISA

MCs (at 1 × 10^6^ cells/mL) in minimal medium were treated with the according inhibitors for 15 min prior to IL-33 addition. Supernatants were collected for cytokine measurements 24 h later. ELISAs were performed according to the manufacturers’ instructions. The following kits were used: IL-8, TNFα (Thermo Fisher Scientific, Berlin, Germany, catalogue numbers: 88-8086-22 and 88-7346-22), CCL1 and CCL2 (DuoSET, R&D Systems, Wiesbaden-Nordenstadt, Germany, catalogue numbers: DY272 and DY279). In addition to the supernatants of stimulated cells, those of unstimulated cells were run (shown in Figure 4b as w/o). As negative controls for each ELISA kit culture medium without cells and dilution buffers were measured on the same plate as the treated samples, routinely giving negative results. For combinations of stimuli, IL-33 was applied for 15 min followed by AER-37, c48/80, or SP for 24 h. To distinguish between whether effects were additive or synergistic, values from single treatments were summed (theoretical outcome in case of an additive effect) and compared to the actual measured values of the according combinations. For determination of the “synergism factor”, values measured in the combined settings were divided by the calculated sums of single stimulations. A value of 1 signifies that the effect was merely additive (calculated = measured); the more the value exceeds 1, the greater the synergism.

### 4.4. Immunoblot Analysis

MCs stimulated with IL-33 were collected by centrifugation and immediately solubilized in SDS-PAGE sample buffer and boiled for 10 min. Inhibitors (p38- and JNK-inhibitor) were applied 15 min prior to the addition of IL-33. Samples corresponding to equal numbers of cells were subjected to immunoblot analysis. The primary antibodies, all purchased from Cell Signaling Technologies (Frankfurt am Main, Germany), were as follows: anti-pp38 (#9211), anti-pSAPK/JNK (#9251), anti-p65 (#8242), anti-pp65 (Ser536) (#3033), anti-p105/50 (#13586), anti-IκB α (#4812), anti-β-Actin (#4967), and anti-Cyclophilin B (#43603), the latter two as loading controls. Cyclophilin B is an ER-specific cyclophilin [119] and a member of the family of peptidyl-prolyl cis-trans isomerases [120]. It is generally used for whole cell/cytosolic lysates. Regarding the presented immunoblot data (Figure 2a and Figure 3a) it was chosen mainly due to its size (20 kDa), thereby not interfering with any other antibody. A peroxidase-conjugated goat anti-rabbit IgG was used as the detection antibody (Merck, #AP132P). Proteins were visualized by a chemiluminescence assay (Weststar Ultra 2.0, Cyanagen, Bologna, Italy) according to the manufacturer’s instructions and bands were recorded on a chemiluminescence imager (Fusion FX7 Spectra, Vilber Lourmat, Eberhardzell, Germany). The quantification of recorded signals was performed using the ImageJ software (Rasband, W.S., ImageJ, U. S. National Institutes of Health, Bethesda, MD, USA, https://imagej.nih.gov/ij/, accessed on 29 March 2021; 1997–2018). Individual intensity values for the detected proteins were normalized to the intensity of the housekeeping protein cyclophilin B of the same membrane.

For translocation experiments, nuclear and cytoplasmic fractions were prepared using the NE-PER buffer system (Thermo Fisher Scientific, Berlin, Germany) according to the manufacturer’s instructions. Protein concentration was determined by the BCA (bicinchoninic acid) method (Thermo Fisher Scientific). Equal protein amounts (~15 µg) of cytoplasmic and nuclear fractions were separated through 4–12% Bis-Tris gels (Thermo Fisher Scientific), transferred to nitrocellulose membranes, and the detection of p65, p105/p50, and beta-actin was performed as described above. Beta-actin was chosen as a loading control due to its presence in the cytosol as well as nucleus [121,122,123,124,125], and as its size does not overlap with the proteins of interest.

### 4.5. Statistics

Statistical analyses were performed using PRISM 8.0 (GraphPad Software, La Jolla, CA, USA). Unless otherwise specified, two-group comparisons were performed using the *t*-test, while for calculations of differences between more than two groups the RM one-way ANOVA with Dunnett’s multiple comparisons test was applied. *p* < 0.05 was considered statistically significant.

## 5. Conclusions

Like other MC subsets, human skin MCs respond to IL-33 by cytokine generation, but their pattern is unique comprising TNF-α, CCL1, CCL2, IL-8, IL-5, and IL-13, but not IL-1β, IL-6, VEGFA, or IL-31. IL-33 activates JNK, but more potently p38 and the NF-κB pathway in cutaneous MCs. The activation of cytokine genes requires NF-κB in the first place, while cytokine release also depends on p38 and, to some extent, JNK, indicating their involvement at the post-transcriptional stage. IL-33 does not only induce cytokines on its own, but it also co-operates with FcεRI and even more so MRGPRX2 in a far more than additive manner to elevate cytokine production. By targeting MCs, IL-33 on its own and more potently when present together with other stimuli therefore creates a cytokine-favoring micromilieu in the skin that likely contributes to inflammation and disease.

## Figures and Tables

**Figure 1 ijms-22-03580-f001:**
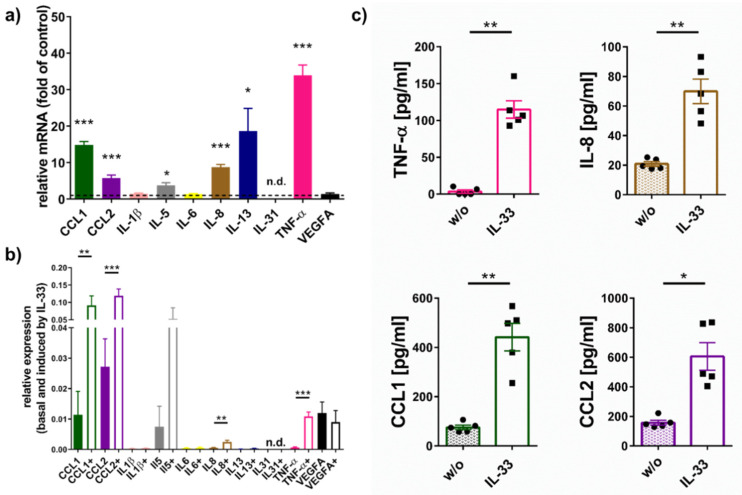
IL-33 triggers several cytokines in human skin mast cells (MCs). (**a**) Skin-derived MCs were stimulated for 90 min with IL-33 or kept in medium alone and harvested for mRNA quantification of different transcripts by RT-qPCR against a total of three housekeeping genes. A one sample *t*-test (against 1) was performed for each cytokine. (**b**) Uses the same dataset as in (**a**), with values normalized to actin as one exemplary housekeeping gene. Basal (plain cytokine name) and IL-33-induced (cytokine name followed by “+”) expression levels of cytokines analyzed by paired *t*-test for each cytokine entity. (**c**) MCs (at 1 × 10^6^ cells/mL) in minimal medium were treated with IL-33 for 24 h when supernatants were collected, and cytokine proteins were quantified by ELISA. Paired *t*-tests were performed for comparison of IL-33 treatment to control (w/o). Each dot represents an independent MC preparation. Data are presented as mean of 5–7 experiments ± SEM. * *p* < 0.05, ** *p* < 0.01, *** *p* < 0.001 n.d. = not detectable.

**Figure 2 ijms-22-03580-f002:**
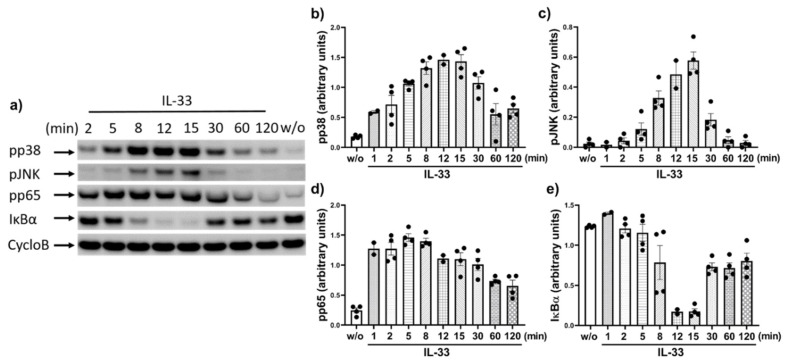
IL-33 elicits pp38, pJKN, pp65, and IκB degradation in human skin MCs. MCs were treated with IL-33 for the times given and the phosphorylation of kinases and components of the canonical NF-κB pathway were detected by immunoblot. (**a**) Representative time-course. (**b**–**e**) Image J based the semi-quantification of the detected signals for pp38 (**b**), pJNK (**c**), pp65 (**d**), and IκBα (**e**). Each dot represents an independent MC preparation. Data are presented as mean ± SEM of up to four experiments for each time point (arbitrary units, calculated as explained in Methods). CycloB = cyclophilin B (loading control).

**Figure 3 ijms-22-03580-f003:**
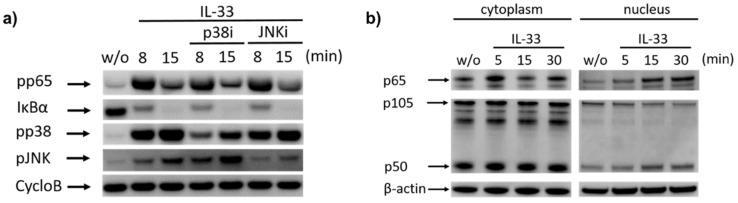
IL-33-mediated activation of NF-κB is independent of p38/JNK and comprises the nuclear translocation of p65. (**a**) To investigate the impact of p38 and JNK on IL-33-induced signaling events, MCs were pre-treated for 15 min with the according inhibitors (or no inhibitor as control) prior to stimulation with IL-33 for 8 and 15 min. Note that neither the p38i nor the JNKi affected pp65 and IκBα, while both inhibitors affected the phosphorylation of their own targets. (**b**) Nuclear and cytoplasmic fractions of IL-33-stimulated MCs were prepared to visualize NF-κB translocation by immunoblot. Representative immunoblots out of two independent experiments each are shown in a and b. CycloB = cyclophilin B (loading control), p38i = p38 inhibitor (SB203580), JNKi = JNK inhibitor (SP600125), w/o = without IL-33 (negative control).

**Figure 4 ijms-22-03580-f004:**
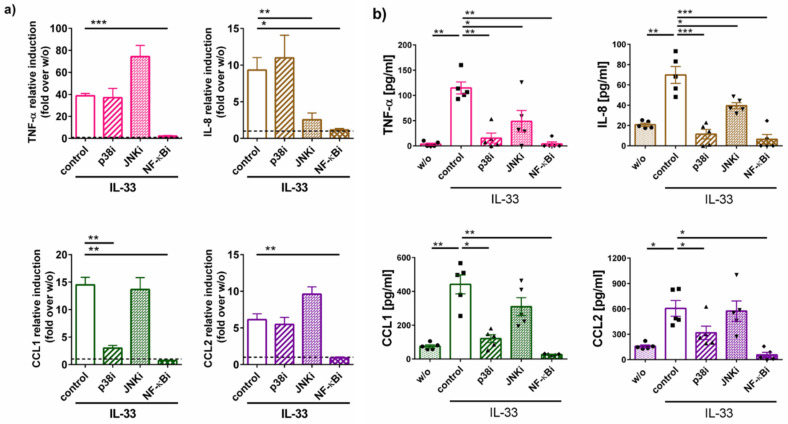
IL-33-induced transcription of cytokine genes chiefly relies on NF-κB, while cytokine release requires p38 activity. (**a**) Cells were pre-treated with the according inhibitors for 15 min (or no inhibitor as control), then stimulated with IL-33 for 90 min. mRNA quantification was performed by RT-qPCR. (**b**) MCs were treated with IL-33 for 24 h when supernatants were collected. Cytokine proteins were quantified by ELISA. Each dot represents an independent MC preparation. Data are presented as mean of 5–7 experiments ± SEM. * *p* < 0.05, ** *p* < 0.01, *** *p* < 0.001, p38i = p38 inhibitor (SB203580), JNKi = JNK inhibitor (SP600125), NF-κBi = NF-κB inhibitor (Bay11-7082), w/o = without IL-33 (negative control).

**Figure 5 ijms-22-03580-f005:**
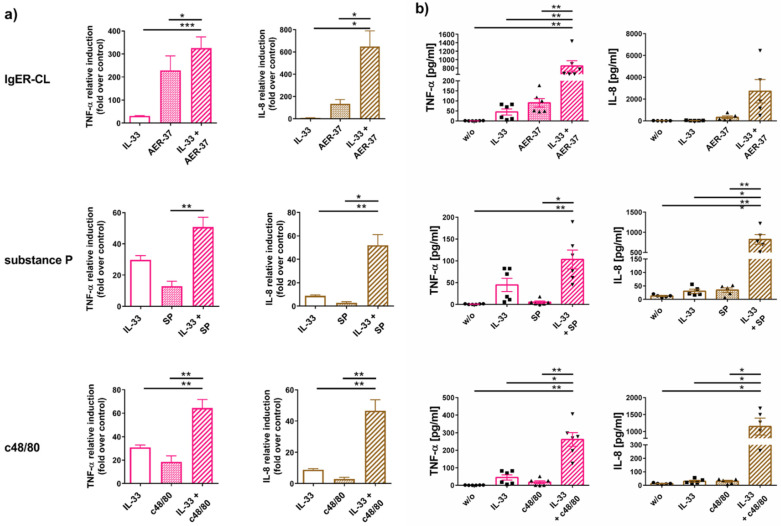
IL-33 synergizes with FcεRI and MRGPRX2 to drive cytokine responses—differences between mRNA and protein. (**a**) MCs were stimulated with IL-33 or AER-37 (FcεRI-CL) or SP or c48/80 (the latter two to activate MRGPRX2) as single treatments for 105 min or pre-treated with IL-33 for 15 min prior to the addition of the other stimuli (combined treatment) for 90 min. TNF-α and IL-8 were quantified by RT-qPCR. (**b**) The same treatments as described under (**a**) were applied, and the supernatants collected after 24 h when TNF-α and IL-8 were quantified by ELISA. Each dot represents an independent MC preparation. The data are presented as mean ± SEM of 5–7 experiments. * *p* < 0.05, ** *p* < 0.01, *** *p* < 0.001.

**Table 1 ijms-22-03580-t001:** Synergism factors for IL-33-induced cytokines (mRNA and protein).

**Cytokine** (**RNA**)	**IgER-CL**	**c48/80**	**SP**
TNF-α	1.10	1.38	1.43
IL-8	5.95	5.20	4.56
CCL1	1.90	4.31	3.24
CCL2	1.47	2.35	2.11
**Cytokine** (**Protein**)	**IgER-CL**	**c48/80**	**SP**
TNF-α	8.12	5.67	2.72
IL-8	7.59	15.71	13.50

**Table 2 ijms-22-03580-t002:** Primer pairs used for RT-PCR.

Gene	Forward 5′-3′	Reverse 5′-3′
TNF-α	TCTCGAACCCCGAGTGACAA	TCAGCCACTGGAGCTGCC
IL-8	ATGACTTCCAAGCTGGCCGTGGCT	TCTCAGCCCTCTTCAAAAACTTCTC
CCL1	TTGCGGAGCAAGAGATTCCC	GGCAGTGCCTCAGCATTTTT
CCL2	CCCCAAGCAGAAGTGGGTTC	TTGGGTTGTGGAGTGAGTGTT
IL-1β	CGAGGGAGAAACTGGCAGAT	AAGCCATCATTTCACTGGCG
IL-5	GAGTCAAACTGTGCAAGGGG	TGGCTGCAACAAACCAGTTT
IL-6	ATGTAGCCGCCCCACACAGA	CATCCATCTTTTTCAGCCAT
IL-13	CATCCGCTCCTCAATCCTCT	GATGCTCCATACCATGCTGC
IL-31	CCCGTCCGTTTACTACGACC	TTGAGATATGCCCGGATGGC
VEGFA	GAAGAAGCAGCCCATGACAG	CTCACACACACACAACAAGG
β-actin	CTGGAACGGTGAAGGTGACA	AAGGGACTTCCTGTAACAATGCA
Cyclophilin B	AAGATGTCCCTGTGCCCTAC	ATGGCAAGCATGTGGTGTTT
GAPDH	ATCTCGCTCCTGGAAGATGG	AGGTCGGAGTCAACGGATTT

## Data Availability

No datasets were generated or analyzed during this study.

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
