# Peer review of "Cytokines Stimulated by IL-33 in Human Skin Mast Cells: Involvement of NF-κB and p38 at Distinct Levels and Potent Co-Operation with FcεRI and MRGPRX2"

_ijms, 2021, doi:10.3390/ijms22073580_

Round 1
Reviewer 1 Report
The presented work is very interesting, but the most significant lack of it is that the authors did not check whether the studied mast cells isolated from the skin had the MRGPRX2 receptor and its functionality. Consequently, the paper has too far-reaching conclusions. Also confusing is the title pointing to a large-scale study that the work does not present. The authors do not describe the receptor's mechanism of action. Looking at the results, I am not sure what the actual MRGPRX2 contribution is. Relying only on cellular responses, it cannot be pointed out.
Generally:
- Show that the indicated mast cells express MRGPRX2 mRNA and protein.
Results:
- Why are IL-8 and CXCL8 not used uniformly but interchangeably in the text?
- Description of Figure 1 – What it means "Each dot represents an independent MC preparation." Is it a pool of mast cells? Or single isolation from one donor? Please clarify.
- Figure 2 b-e – Are the colors presented in this section are dictated by something specific? 30' it is almost invisible. Please clarify.
- Figure 3 a, b – Please clarify the selection of these two loading controls and inconsistent time sequences.
Materials and Methods:
- Please indicate how the purity of the isolated mast cells was assessed. (line 397)
- How were the concentrations of the inhibitors used for a given research model selected? (line 408)
- Please indicate the sensitivity of the protein assay kits (line 397). What controls Authors used in ELISA? Please add to the Materials and methods section.
Discussion:
- Please clarify PMCs (line 293).
Reviewer 2 Report
The authors described here inflammatory cytokines production in human cutaneous mast cells induced by IL-33. Accumulating evidence suggests that IL-33 is one of the critical mediators that could modulate mast cell activation. Because MRGPRX2-mediated induction of inflammatory cytokines remains largely unknown, their findings of synergistic effects of IL-33 on MRGPRX2-mediated cytokine production might be of great significance in the field of mast cell research. Several concerns described below should be addressed before publication.
The descriptions such as "one cytokine activates the other cytokines" in the abstract should be revised, because they vaguely describe the results.
When the expression of a certain gene is negligible, it might be inappropriate to use "fold of change", because the standard expression level is not measurable. The authors should compare the mRNA expression levels in the cells treated with or without IL-33 using the loading control. In principle, one could not compare the mRNA expression levels among the different genes because the efficiencies of the PCR amplification using the different primer pairs vary.
Is Cyclophilin B a good loading control?
In Figure 2a, the obvious band shift appears in the later phase in the blot obtained using anti-phosphorylated p65. The authors should discuss it.
Because p38 and JNK are phosphorylated respectively by their specific MAPKK, not autophosphorylated, the authors should explain it why the inhibitors of p38 and JNK could affect their phosphorylated statuses.
The nuclear extraction should be verified using the antibodies raised against the nuclear fraction-speciifc proteins. In Figure 3b, both fractions contain the similar levels of beta-actin.
The authors should briefly present the procedures of preparations of human foreskin-derived cultured mast cells, because it might be of great importance for the readers to recognize the type of mast cells used in this study.
Round 2
Reviewer 1 Report
Thank you, all my comments have been applied.